



# Frontal collapse of San Quintín glacier (Northern Patagonia Icefield), the last piedmont glacier lobe in the Andes

Michał Pętlicki[1,2], Andrés Rivera[3], Jonathan Oberreuter[4], José Uribe[4], Johannes Reinthaler[5], and Francisca Bown[6]

[1]Faculty of Geography and Geology, Jagiellonian University, Cracow, Poland
[2]Departmento de Geografia, Universidad de Concepcion, Concepcion, Chile
[3]Departmento de Geografia, Universidad de Chile, Santiago, Chile
[4]Centro de Estudios Cientificos, Valdivia, Chile
[5]University of Zurich, Zurich, Switzerland
[6]Laboratorio de Glaciologia, TamboAustral Geoscience Consultants, Valdivia, Chile

**Correspondence:** Michał Pętlicki (michal.petlicki@uj.edu.pl)

**Abstract.** Glacier fronts are retreating across the globe in response to climate warming, revealing valleys, fjords, and proglacial lakes. The piedmont lobe of San Quintín, the largest glacier of the Northern Patagonia Icefield, in southern Chile, has recently entered a phase of frontal retreat, where its terminus is rapidly disintegrating into large tabular icebergs calving into a new proglacial lake. We present results of a new airborne GPR survey of the terminus of this large Patagonian glacier (763 km² in 2017), complemented with an analysis of ice flow velocity, satellite imagery, and ice elevation change to show that the ongoing retreat is caused by recent detachment of a floating terminus from the glacier bed and may shortly lead to the disappearance of the last existing piedmont lobe in Patagonia. Finally, we discuss how the observations of San Quintín's ongoing collapse may give insights into processes governing frontal retreat of fast-flowing temperate glaciers and the quasi-stability of the floating termini.

## 1 Introduction

A piedmont glacier is a unique type of glacier flowing out of a mountain range, spilling to all directions once it is free of constraining valley walls, forming a circular lobe at low elevation. To develop a piedmont lobe, a specific combination of factors is required. First, it needs a topographic configuration favouring a glacier to flow out from a mountain range reaching lowlands where there is non-lateral topographic constrain for the ice, that debouches or expand laterally forming a lobe or piedmont. Second, a suitable geological substrate is needed, with soft sediments deposited at the foot of the mountains, that allow relatively high values of bottom slip, enhancing flow in the terminal zone. Generally, the presence of a piedmont glacier can be considered a sensitive indicator of cold climate conditions that are favourable to glacier growth and development. Unfortunately, glaciers around the world are currently retreating on an unprecedented scale due to anthropogenic climate change (Zemp et al., 2015), and only a few piedmont glaciers in the world have survived. The remaining and most recognised examples are the Alaskan glaciers, such as the Malaspina lobe (Muskett et al., 2008), and the glaciers of the Canadian Arctic (Alex Heiberg Island). However, in the southern hemisphere outside Antarctica, piedmont glaciers are nowadays difficult to



find, even though they were much more common in the past, especially during the Last Glacial Maximum when the Patagonia Ice Sheet developed a large piedmont glacier that ran down the mountains into the Chiloe area, or further north in the Chilean Lake District where many smaller piedmont lobes upon retreat formed large lakes such as Lago Ranco, Llanquihue, or Villarrica (Bentley, 1997). Further south in Patagonia, the piedmont glaciers prevailed much longer (Davies et al., 2020). A perfect

example is the San Rafael glacier, which formed a large piedmont lobe at the beginning of the last century before it eventually collapsed and the front retreated into the valley where it now slowly retreats inland (Koppes et al., 2010).

Currently, the last piedmont glacier in the Andes is the San Quintín glacier, which flows from the Northern Patagonia Icefield (NPI) plateau to the Istmo de Ofqui, creating a vast lobe (Fig. 1). The NPI (Figure 1) is the second largest ice mass in South America with a surface area of about 4000 km$^2$ (Aniya, 1988; Rivera et al., 2007; Windnagel et al., 2022). Most of its outlet

glaciers are subject to pronounced frontal retreat (Rivera et al., 2007; Willis et al., 2012); however, there are some examples of relatively stable front positions in recent decades, the most prominent being the San Rafael Glacier (Mouginot and Rignot, 2015; Collao-Barrios et al., 2018). In general, the NPI experiences a net negative surface mass balance (Schaefer et al., 2013) that is reflected in a consistent lowering of the ice surface (Willis et al., 2012; Abdel Jaber et al., 2019; Dussaillant et al., 2019) in recent decades. Consequently, NPI is an important net contributor to the observed rise in sea level (Rignot, 2003; Braun

et al., 2019).

The largest outlet glacier of NPI, San Quintín (763.3 km$^2$ in 2019; DGA, 2022), calves into an unnamed freshwater lagoon that has limited river outflow, not allowing the discharge of icebergs to the ocean (Podgórski and Pętlicki, 2020). The glacier flows to the western side of the NPI and is therefore exposed to the extremely wet and temperate maritime climate (Lenaerts et al., 2014) that causes rapid thinning (Braun et al., 2019) due to the negative ice mass balance with ablation rates exceeding

10 m·a$^{-1}$ (Schaefer et al., 2013) or dynamic stretching. Interestingly, during his famous expedition to Patagonia at the beginning of the twentieth century, Steffen (1900) described this glacier as much larger than the neighbouring San Rafael glacier. The Swedish expedition of Otto Nordenskiold mapped the glacier extent in 1920-21, and by that time the piedmont lobe extended further than the current lagoon (Fig. 1). Since then, due to its challenging access and logistics, San Quintín has been the subject of only a few detailed studies, most of which concerned its terminal moraines (Winchester and Harrison, 1996; Harrison et al.,

2001), or were based mainly on remote sensing data (Podgórski and Pętlicki, 2020). Consequently, large-scale studies of the ice elevation change (Rignot, 2003; Rivera et al., 2007; Braun et al., 2019; Abdel Jaber et al., 2019), ice thickness (Gourlet et al., 2016; Millan et al., 2019), or glacier mass balance (Schaefer et al., 2013; Bravo et al., 2021; Minowa et al., 2021) of NPI suffer from having little constraint from *in situ* measurements on one of its largest glaciers. More generally, due to sparse measurements, there is an urgent need for more observational data on most glacial and climate variables (Ruiz et al., 2022), and

especially on the thickness in the frontal parts of the Patagonian outlet glaciers (Millan et al., 2019) as the currently available data are restricted to only a few examples and the results of inverse modelling are in many cases inconsistent (Carrivick et al., 2016; Farinotti et al., 2019). Filling this gap may have considerable implications for the calculation of mass flux and the partition between frontal ablation and surface mass balance, as both depend strongly on the front geometry (Bown et al., 2019; Minowa et al., 2021).



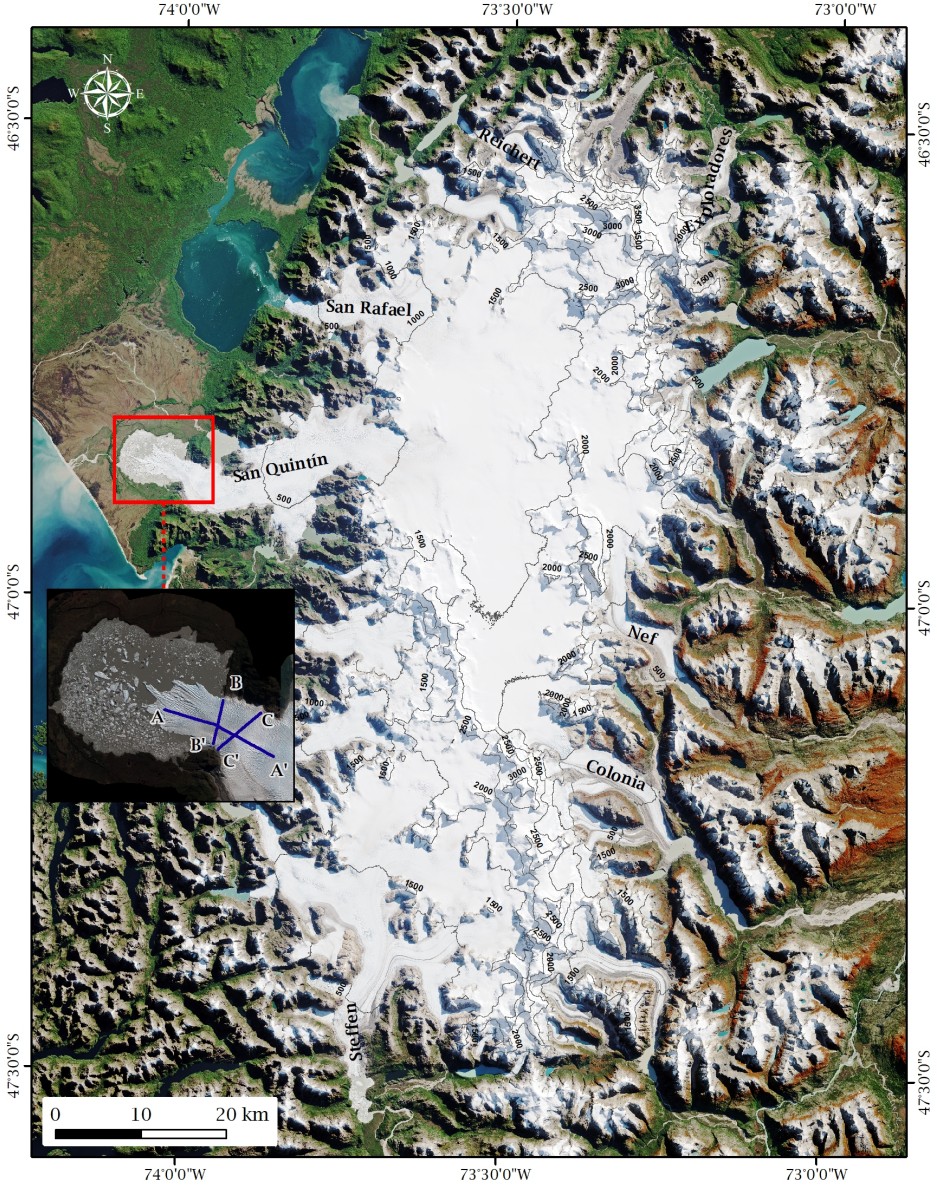

**Figure 1.** Map of the study area: (a) general location with a map of the Northern Patagonia Icefield, (b) San Quintín glacier with marked GPR profiles in the terminal part. Background imagery: Sentinel-2A acquired on 16/04/2017.

Usually, the position of the grounding line is determined with indirect methods, mostly employing SAR interferometry or flow modelling. The few direct studies focused on the Antarctic shelf glaciers and the Greenland outlet glaciers (MacGregor et al., 2011; Mayer et al., 2018; Rosenau et al., 2013). Furthermore, there is a lack of direct observations of a collapse of a piedmont glacier; most studies are based on the reconstruction of past changes in the area of such glaciers (i.e, Lago Ranco;



Bentley, 1997). When a glacier detaches from the bed and becomes afloat, the balance of forces and stresses changes (Tsutaki et al., 2019; Pronk et al., 2021). So far, it is not clear how long it takes for a floating tongue to collapse, that is, how many years after it becomes buoyant, it fragments and disintegrates (Sutherland et al., 2020). At the same time, the process of calving has already been studied in the case of glaciers flowing into lakes (Boyce et al., 2007), especially in Patagonia (Warren et al.,

2001) and New Zealand (Warren and Kirkbride, 2003). However, these studies do not provide an exact timing for the glaciers reaching flotation.

To identify whether the San Quintín glacier front is afloat and to determine its ice thickness and grounding line position, a helicopter-borne radar sounding was carried out in May 2019. The results of this survey allowed us to determine the strength of the reflection from the glacier bed and, thus, the presence of water under the glacier's terminus. Interestingly, so far, no similar

research has been carried out on other Patagonian glaciers.

## 2  Data and Methods

### 2.1  Satellite imagery, glacier extent and elevation change, ice flow velocity

We calculate the elevation change of the ice extent based on satellite Digital Elevation Models (DEMs) from 2017 and 2019 constructed with Very High Resolution (VHR) satellite imagery - Worldview-2 (2017) and Pléiades (2019) and complemented

with the Shuttle Radar Topography Mission (SRTM) DEM from 2000 and TanDEM-X DEM from 2013-2015. VHR images were processed with ERDAS IMAGINE, and corresponding 2 m resolution DEMs were produced. More details on DEM generation can be found in the Appendix A.

The glacier extent was also manually mapped with the GEEDiT tool (Lea, 2018) on a set of available optical imagery (Landsat 4-5, 7, 8 and Sentinel-2 covering the period from 1986 to 2020. The high sediment content of lake water and on some

icebergs, and the persistent presence of ice melange prevented the efficient use of automated methods for the delineation of the glacier front (Podgórski and Pętlicki, 2020).

Ice flow velocities are extracted from two publicly available global datasets: the FAU RETREAT database of monthly mosaics for 2015-2021, based on Sentinel-1 SAR imagery cross-correlation (Friedl et al., 2021), and from the NASA MEaSUREs ITS_LIVE project (Gardner et al., 2019), where yearly mosaics are provided based on optical LANDSAT imagery processed

with the auto-RIFT algorithm (Gardner et al., 2018) and covering period 1986-2018. Taken together, they cover the entire study period of 1986-2021 with gaps with no data in 1987-1998.

### 2.2  Airborne GPR

In May 2019, a helicopter-borne Ground Penetrating Radar (GPR) field campaign was conducted on the terminal part of San Quintín glacier, during calm and cold weather conditions. We used an in-house GPR system with a high-voltage impulse

transmitter (3.2 kV peak output, with a rise time pulse of 8 ns), a signal acquisition receiver, a laser altimeter, and a dual frequency GPS (Zamora et al., 2009, 2017). A dual broadband bow-tie antenna operating at a central frequency of 20 MHz was



contained within a metal structure of 8.0 x 4.8 x 1.2 m ($\sim$350 kg) hanging 20 m below the helicopter during measurement. The received signal was digitised by a 14-bit analogue-to-digital converter at a sampling frequency of 400 MHz. During the survey, the GPR was set at 3 kHz of the pulse repetition frequency with a stacking of 256 traces. With this configuration and considering a flight speed of 20 m·s$^{-1}$, a radar trace was obtained for each $\sim$2 m, with a vertical resolution of $\sim$5 m. This system is capable

of penetrating up to 700 m of temperate ice in Patagonia (Millan et al., 2019). Radar data were geo-located with differential GPS, using the GPR GPS receiver and a GPS base station located in Puerto Guadal, approx. 80 km east of the study site. The two-way travel times and amplitudes of the radar reflectors were determined using Reflexw software (http://www.sandmeier-geo.de/), and the ice thickness was calculated assuming an electromagnetic wave velocity propagation in ice of 0.168 m·ns$^{-1}$. The surface and bottom reflectors were manually selected. A static correction, band-pass filter, background removal, equidistant

trace interpolation, 2D diffraction migration, and a gain function were applied to the raw data. The elevation of the glacier surface was estimated by subtracting the radar distance between the antenna and the surface from the GPS elevation of the antenna, using an airwave speed of 0.3 m·ns$^{-1}$. The errors in estimating ice thickness came mainly from the assumption of electromagnetic wave velocity, which depends on snow and ice density. Together with other uncertainties such as GPS position error, radar resolution, and picking error, we estimate the overall uncertainty at $\sim$6% of ice thickness.

## 2.2.1  Bedrock Reflective Power

The depth to the bedrock and the presence of water at the glacier bed were classified on the basis of the strength of the reflection signal. In order to quantify the properties of the reflection from the glacier bed, we use the Bedrock Reflective Power (BRP). Given a radar profile and the vectorisation of the air–ice and ice–bed interface, the BRP was calculated according to Gades et al. (2000):

$$BRP = \frac{1}{n_2 - n_1 + 1} \sum_{n=n_1}^{n_2} S_n^2 \qquad (1)$$

where $S_n$ is the amplitude of the radar signal in each sample; $n_1$ and $n_2$ are the initial and final sample numbers of the time window that includes the background reflector. The values $n_1$ and $n_2$ were chosen following Copland and Sharp (2001).

The Internal Reflection Power (IRP) is a measure of the radar energy dissipated and reflected within the ice column and is given by the following:

$$IRP = \sum_{n=n_1}^{n_2} S_n^2 \qquad (2)$$

In this case, the Copland and Sharp (2001) guidelines were again followed for the choice of $n_1$ and $n_2$.

The traces with ice thickness lower than 600 m were filtered and removed from the BRP analysis. The estimate of BRP from Equation 1 depends on the attenuation by geometric divergence and the dielectric properties of the ice. Therefore, an estimated



BRP ($BRP_e$) is calculated to compensate for these effects and finally obtain a $BRP_R$ (Residual Bed Reflection Power) which is a measure of bed properties. $BRP_R$ is defined as:

$$BRP_R = \frac{BRP}{BRP_e} \tag{3}$$

To determine $BRP_e$, we start with the radar equation:

$$P_{rx} = P_{tx}GL_gT_{12}^2L_i^2R_{23} \tag{4}$$

where $P_{rx}$ is the power received, $P_{tx}$ is the transmitted power (76.12 dBm), $G$ is the total combination of gains and losses of the radar system (5.3 dBi), $L_g$ is the geometric divergence loss, $T_{12}$ is the transmission loss at the air-ice interface, $L_i$ is the dielectric loss in the ice column and $R_{23}$ is the background reflection coefficient. Thus, we can write

$$BRP_e = P_{tx}GL_gT_{12}^2L_i^2 \tag{5}$$

The loss due to geometric divergence in the case of aerial measurements (Gacitúa et al., 2015) is given by the following:

$$L_g = \frac{(G\lambda)^2}{\left[8\pi\left(h_a + \frac{h_i}{\sqrt{\varepsilon_i}}\right)\right]^2} \tag{6}$$

where $\lambda$ is the wavelength in the air, $\varepsilon_i$ is the relative dielectric permittivity of the ice, $h_i$ is the thickness of the ice, and $h_a$ is the thickness of the air layer. The dielectric loss in ice is obtained with the expression (Jacobel et al., 2009):

$$L_i = e^{-2h/L_a} \tag{7}$$

where $L_a$ is the average length of attenuation, which is related to the average attenuation rate $N_a$ (in dB·km⁻¹) through the following relationship:

$$N_a = 10^3\left(10\log_{10}e\right)L_a^{-1} \tag{8}$$

This expression was obtained by linearising the equation 4 after normalising with the term $L_g$. Thus, the attenuation of the ice $N_a$ (in dB·km⁻¹) was calculated using the empirical method (Jacobel et al., 2009; Hills et al., 2020), through a linear
regression based on the decay of the amplitude with the depth of the ice. For the linear regression, the linear range of the dispersion was considered from 200 to 600 m in ice thickness. The results of the attenuation (depth-averaged attenuation) are shown in Figure 2. According to a compilation of several studies by Hills et al. (2020), $N_a$ varies between 5 and 30 dB·km⁻¹




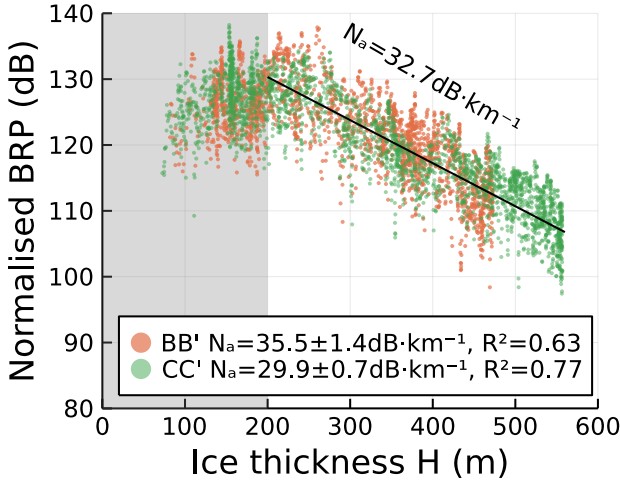

**Figure 2.** Scatter plot of the normalised Bedrock Reflection Power (BRP) vs. ice thickness along the transverse profiles BB' and CC'. The grey shaded area corresponds to ice thickness lower than 200m where the linear relationship does not hold.

in cold ice. The obtained values are slightly higher, which could be expected for temperate ice, which has a higher attenuation compared to cold ice as a result of the higher liquid water content.

The measured Bedrock Reflection Power follows a linear relationship with the ice thickness along all measured profiles (Fig. 2). This allows for elimination of geometric factors and calculation of $BRP_r$ that is related only to the properties of the
glacier bed. With the estimated $N_a$ of 32.7 dB·km$^{-1}$, $BRP_e$ and $BRP_r$ were calculated. The gain $G$ of Equation 6 was adjusted separately for each profile so that the mean $BRP$ is not lower than the mean $BRP_e$ and was as follows: $G_{AA'}$ = 97.5349, $G_{BB'}$ = 97.8349 $G_{CC'}$ = 100.7349.

## 2.3   Ice buoyancy

For grounded glaciers, ice surface elevation change $\Delta Z$ is equivalent to ice thickness change $\Delta H$. However, when the ice is not
supported at the base but is floating, one also has to consider the elevation change of the glacier bed that compensates a large share of the ice thickness change due to buoyancy. Consequently, the ice elevation change of an ice shelf or a floating tongue is typically an order of magnitude lower than that of a grounded tidewater glacier that is exposed to the same ice thickness change. First, let us introduce the flotation criterion, $f$, which depends on the density of both the submerged material and the surrounding fluid:

$$f = \frac{\rho_i}{\rho_w} \tag{9}$$



Following Schwikowski et al. (2013) measurements at the Pío XI glacier, we assumed an ice density $\rho_i$ of 900 kg·m$^{-3}$. Taking the fresh water density $\rho_w$ of 1000 kg·m$^{-3}$, we set $f$ at 0.90. The relative height of the water $\omega$ is a measure of a portion of the cliff that is below the water level:

$$\omega = \frac{h_w}{H} = 1 - \frac{h}{H} \tag{10}$$

where $h_w$ is the thickness of the ice below the water level, $H = h + h_w$ is the total thickness of the ice and $h$ is the thickness of the ice above the water level. When $\omega$ approaches $f$ (0.90), the ice reaches flotation. For lower $\omega$ values, it is useful to consider the height above flotation $h_f$ as a measure of the stability of grounding:

$$h_f = H - \frac{h_w}{f} \tag{11}$$

This value tells us how much ice supports the ice column before it reaches flotation. For tidewater glaciers, $\omega$ and $h_w$ have been linked to the intensity of calving (Mercenier et al., 2018; Bown et al., 2019). In the case of lacustrine calving, the ice front typically becomes unstable when it reaches flotation and, consequently, disintegrates to large tabular icebergs after full thickness calving (Warren et al., 2001; Carrivick et al., 2020). If we know the height above the flotation and assume a constant ice thinning rate $\dot{H}$, the time to reach the flotation $t_f$ can be calculated:

$$t_f = \frac{h_f}{\dot{H}} \tag{12}$$

## 3 Results

### 3.1 Ice extent and recent elevation change

Following a slight advance between 1986 and 1993 (Winchester and Harrison, 1996), the San Quintín front has retreated over 3 km until 2020, leaving behind a large lagoon that surrounds the remnants of its piedmont lobe (Fig. 3a). The retreat has neither been continuous nor uniform in space, with a prominent local retreat in particular regions and a more stable position in others. Until 1991, the glacier terminus was positioned near the current border of the proglacial lake, and the lobe had a circular shape, typical of a piedmont glacier. Then until 1999 the northern part of the front retreated, while the southern part remained stationary, and both lateral margins retreated as well. The southern half of the front remained close to the lagoon shore until as late as 2008, whereas other sections of the lobe followed more complicated cycles of retreat and advance with a longer period of stabilisation 2005-2015 approximately 2km upstream from the 1986 position. There were spatially restricted regions of the front that advanced significantly before a collapse, forming "fingers" of tabular shape. Such a phenomenon could be observed in the years 2017-2018 south of the central line. After this, the front retreated and in 2019 it was located approx. 5 km from its 1986 position, and has become much more compact.



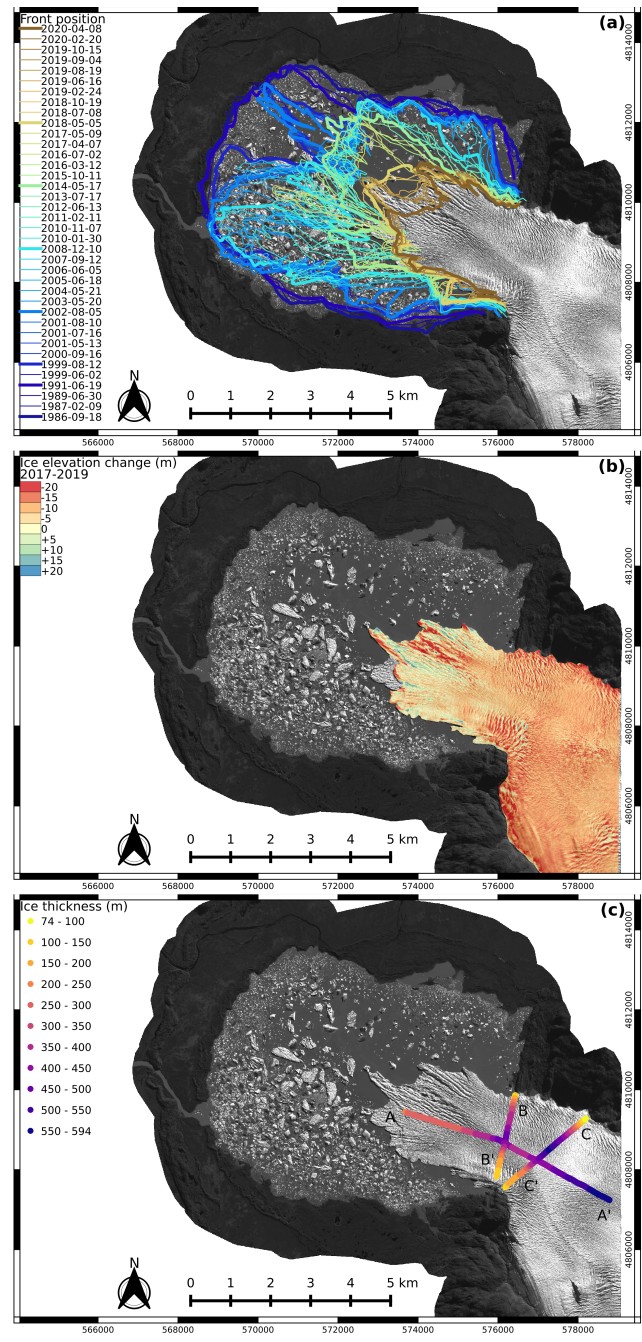

**Figure 3.** (a) Ice front position change of San Quintín Glacier between 1986 and 2020, (b) Ice elevation change of San Quintín Glacier between 2017 and 2019, (c) Ice thickness profiles measured on 19 May 2019. Image background: Pléiades, 24 Apr 2019 (Pléiades © CNES 2019, Distribution Airbus D&S).



The terminal part of San Quintín glacier has overall decreased its elevation substantially during 2017-2019 (Fig. 3b), with the maximum thinning of more than 10 m observed ∼5 km from the 2019 front position and a substantially lower change near the terminus itself, resulting in a negative altitudinal gradient of ice loss. The ice elevation change of the first kilometre from the front position is only slightly negative, with high variability due to changes in crevasse positioning.

## 3.2 Ice thickness and bed properties

The radio echo sounding results show an ice thickness of 200 m at the glacier front and with very little increase over a distance of 1.5 km up-glacier along the longitudinal profile AA'. Further along the profile, the depth to the bed of the glacier increases substantially, to 600 m of thickness 6 km upstream from the 2019 terminus position (Fig. 3c and 4a).

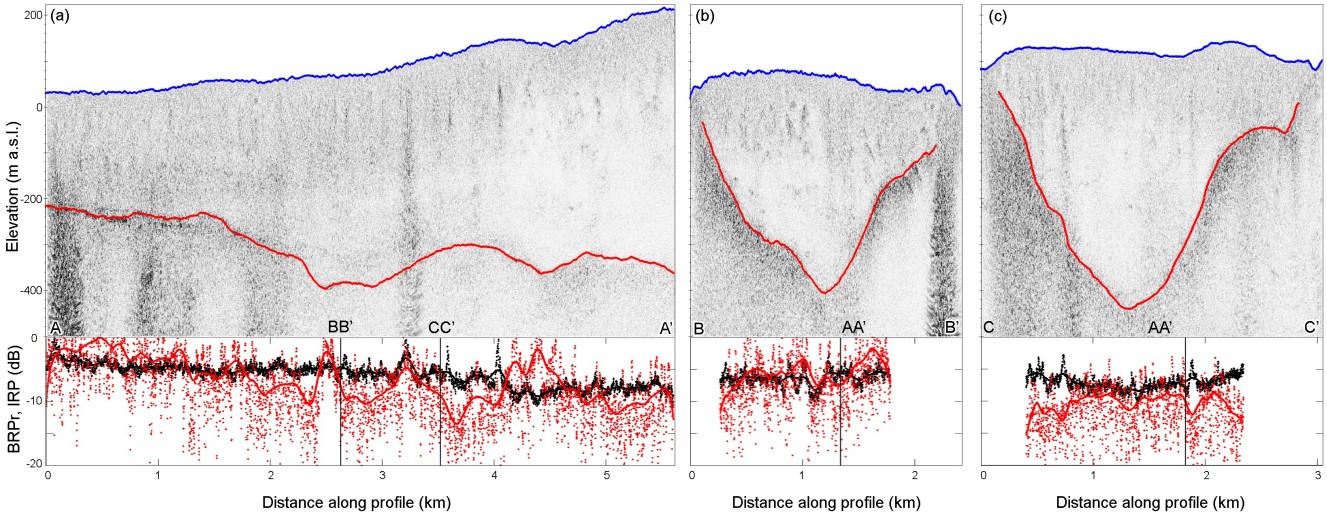

**Figure 4.** GPR profiles of San Quintín glacier measured on 19 May 2019: radargrams (upper panels), normalised relative Bedrock Reflection Power (BRPr) and Internal Reflection Power (IRP) (lower panels) along the longitudinal profile AA' (a), and transverse profiles BB' (b) and CC' (c).

The glacier bed has a reverse slope along the longitudinal profile AA' with a minimum bed elevation of c. 400 m below sea level (Fig. 4a). It follows a parabolic, slightly skewed shape along the transverse profiles BB' and CC' (Fig. 4b and c) located in the confined valley. There is a high scattering in the ice body, typical of temperate Patagonian glaciers. BRP$_r$ along the longitudinal GPR profile shows great variability, with the highest values near the front, coinciding with regions of the lowest ice thickness and a decrease in the elevation of the ice surface (Fig. 4). There are isolated regions with high BRP$_r$ in glacier bed overdeepings near km 2.5 and 4.3. The IRP is generally uniform in space, with isolated regions with higher values that correspond to highly crevassed regions or water inclusions, reaching highest values near 2.4 and 3.2 km of the profile AA'.

### 3.3 Ice buoyancy

There is a clear relationship between ice elevation, $BRP_r$, ice elevation change rate, and relative water height (Fig. 5). When the ice reaches flotation (grey area in Fig. 5a), $BRP_r$ reaches its maximum and the ice elevation change rate goes to zero. In the low elevation region, the relative water height is close to 0.9 and the ice elevation change rate is low, while higher up the glacier, the relative water height decreases with increasing ice elevation change (Fig. 5b). Taken together, this indicates the presence of a floating tongue, where the presence of liquid water at the bed manifests itself in a high $BRP_r$, and the ice buoyancy limits the effect of ice thickness change on ice surface elevation change.

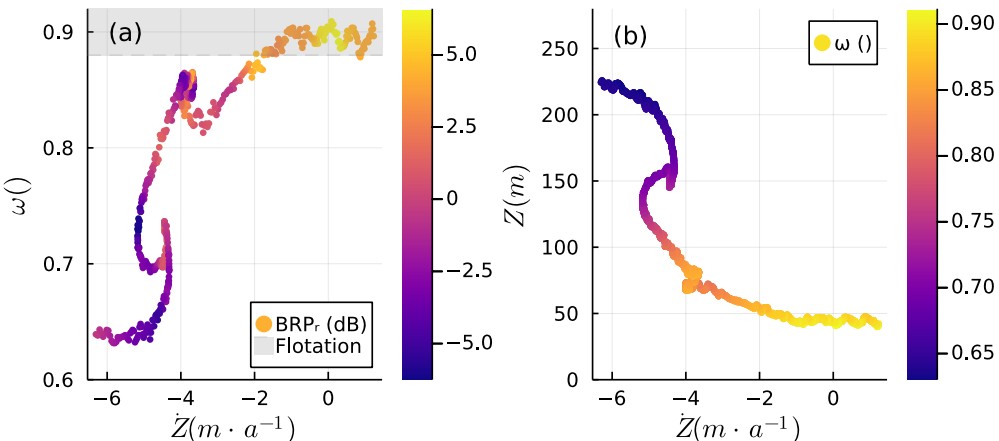

**Figure 5.** The longitudinal profile AA': (a) a scatter plot of relative water height ($\omega$) and ice elevation change rate ($\dot{Z}$) with $BRP_r$ marked with colour; (b) altitudinal profile ($Z$) of ice elevation change rate ($\dot{Z}$) with the relative water height ($\omega$) marked with a colour scale. The grey area in panel (a) corresponds to $\omega$ values typical for floating conditions (0.90±0.02).

### 3.4 Ice velocity

Ice flow velocities (Fig. 6a) show an increase in glacier flow after 2012, especially in the first 6 km up-glacier from the 1986 front position. Seasonal fluctuations in flow (Friedl et al., 2021) are pronounced, with a 25% increase in summer that is up to 300 m·a⁻¹ faster than the winter flow (Fig. 6b). The ice flow velocities derived from both ITS_LIVE and RETREAT FAU datasets are consistent in the overlapping period 2014-2018 (Fig 6b).

Compared to 1986, a general increase in ice flow velocity over time can be observed, although this was interrupted in 2010 when the flow decreased to 400 m·a⁻¹ to later reach its maximum in 2014. Whereas in 1986 and early 2000 the velocity near the terminus was much lower than further upstream, this condition eventually changed after 2010, with smaller terminal speed-up episodes in 2004 and 2006 (Fig. 6a). This suggests that the stagnant ice at the front was mobilised after the terminus detached from the terminal moraines and land and developed a proglacial lake.



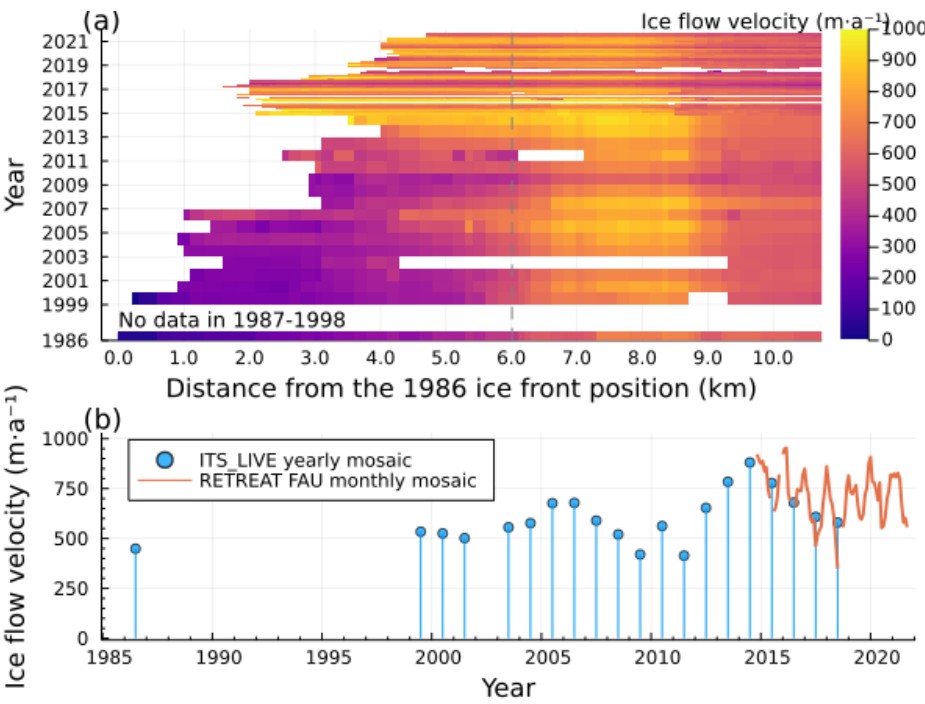

**Figure 6.** Ice flow velocities of San Quintín in 1986-2021: (a) along the profile AA' (b) point values at the distance of 6.0 km from the 1986 ice front position. ITS_LIVE yearly mosaic for 1985-2018 (Gardner et al., 2019) and RETREAT FAU monthly mosaics for 2015-2021 (Friedl et al., 2021).

### 3.5   Glacier change in 2000-2019 along the longitudinal profile

The longitudinal profile of the terminal part of San Quintín Glacier shows irregular bedrock undulations, a significant ice surface lowering, and frontal retreat between 2000 and 2019 (Fig. 7a). To visualise the shape of the proglacial lake, we assumed a parabolic profile with superimposed small sinusoidal oscillations, although it is important to point out that the actual depth

5   of the lagoon remains unknown and therefore it is marked with interrogation signs in the profile.

$BRP_r$ is positive between km 5.2 and 6.5 of the profile, where the surface topography is flat near the front and in regions coinciding with local bed over-deepenings (Fig. 7b). The calculated relative water height $\omega$ reaches $f = 0.9$ in the terminal area for the surface topography of 2019 and the former of 2011/2014, corresponding to the TanDEM-X DEM (Fig. 7c). The values of $\omega$ calculated for the 2000 surface topography (SRTM DEM) are lower and do not reach $f$ for the entire profile. There is a

10   consistent increase in $\omega$ values between 2000, 2011/14, and 2019 along the longitudinal profile upstream of km 6, whereas in the region closest to the terminus (km 5.2 - 6.0), $\omega$ does not increase after 2011/14 as it has already reached its maximum values of 0.9. The second local maximum of $\omega$ is located near km 7 of the profile, in the region of a bed overdeepening (Fig. 7a). In this area, $\omega$ increases from 0.73 in 2000, to 0.81 in 2011/14, finally reaching 0.86 in 2019 (Fig. 7c). Higher up-glacier, $\omega$



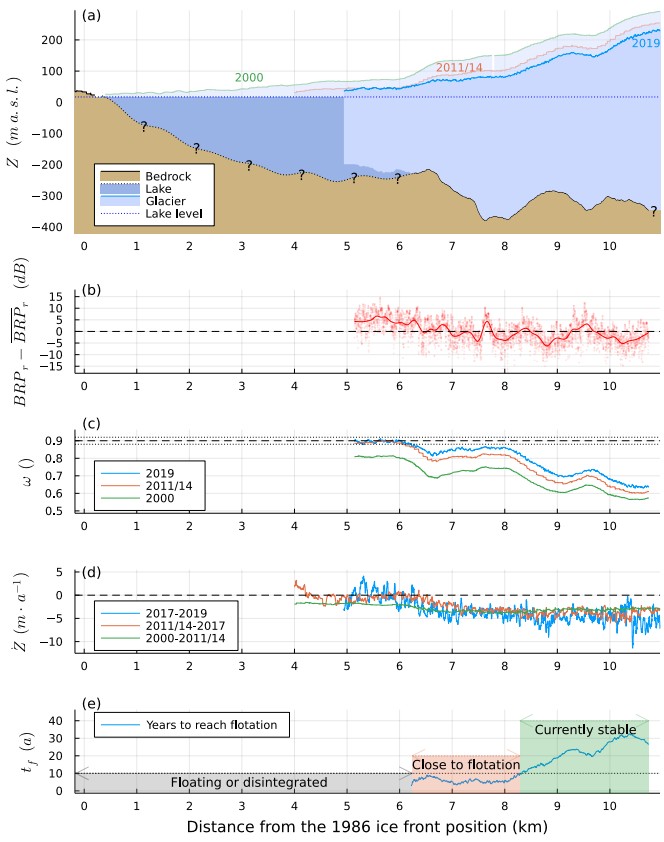

**Figure 7.** Longitudinal profile of San Quintín: (a) glacier geometry in 2000-2019; (b) relative Bedrock Reflection Power (BRPr) anomaly; (c) relative water depth and flotation criterion; (d) ice elevation change rate in 200-2019; (e) time to reach flotation if the 2017-2019 ice elevation change rate remains constant. The exact location of the longitudinal profile is shown in Figure 8.

gradually decreases with small local maxima at km 9.7, finally reaching values of 0.58 in 2000, 0.61 in 2011/14, and 0.63 in 2019 at the end of the longitudinal profile.

Throughout the observation period 2000-2019, there is an inverted profile of the ice elevation change rate $\dot{Z}$, with more negative values up-glacier and a smaller lowering in the frontal region of the glacier (Fig. 7d). Between 2000-2011/14 (SRTM and DEM-TanDEM-X) and 2011/14-2017 (TanDEM-X DEM and WV-2 DEM), there was a significant decrease in the absolute value of $\dot{Z}$ in the frontal region of the glacier, between km 4 and 7, while higher up-glacier the changes were small. On the contrary, the change in $\dot{Z}$ between 2011/14-2017 and 2017-2019 (WV-2 and Pléiades DEMs) near the terminus (km 5.0-6.5) was very small, while there was a larger change near km 6.5-7.0. Furthermore, there is significant noise in 2017-2019 $\dot{Z}$, related to the high resolution of the compared DEMs that capture very small surface undulations due to crevassing.

Assuming that the ice elevation change rate $\dot{Z}$ is constant in time (Fig. 7d), the calculated values of the time to reach flotation $t_f$ are presented in Figure 7e. Low values of $t_f$ spatially correspond to the regions of the longitudinal profile where $\omega$ is low,

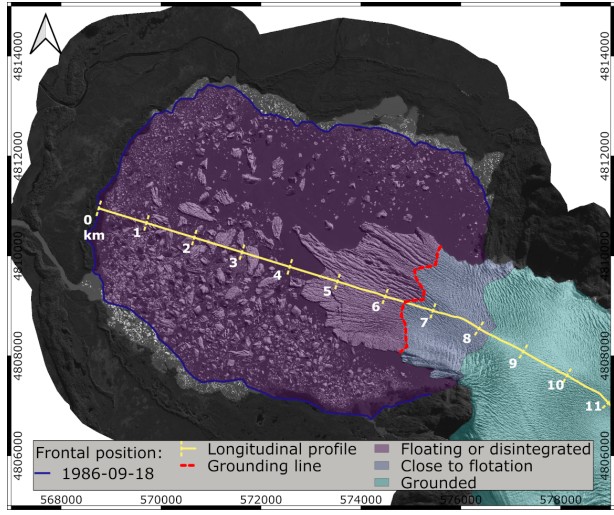

**Figure 8.** Location of the longitudinal profile, the grounding line and classification of the glacier stability regions in May 2019. Grounding line location is exact at its intersection with the longitudinal GPR profile, in other places it is interpreted from the ice elevation change field and crevassing patterns, and should be regarded as indicative. Image background: Pléiades, 24 Apr 2019 (Pléiades © CNES 2019, Distribution Airbus D&S).

i.e. where the ice is close to flotation. We assign a threshold of $t_f$=10 years to distinguish regions that can be considered stable, and those with lower values of $t_f$ are prone to disintegration in the near future. There is a sharp increase in $t_f$ from km 8.0 to km 9.3 of the longitudinal profile, and the glacier section from 6.2 to 8.2 km along the profile is close to flotation and can destabilise in the coming decade (Fig. 8).

## 3.6 Comparison with other ice thickness estimates

The measured ice thickness is lower than the estimates of Carrivick et al. (2016) and Millan et al. (2019), and higher than Farinotti et al. (2019), none of the previous data sets is accurately approximating the geometry of the terminal part of San Quintín as mapped here (Fig. 9). The correlation coefficients of modelled vs. measured ice thickness are very low, and the bias is very high. The Carrivick et al. (2016) data set overestimates the thickness of the ice by 100%, although the differences show a linear relationship with depth ($R^2$=0.55). On the contrary, the Farinotti et al. (2019) data set shows very little correlation with the measured values ($R^2$=0.07) and in general underestimates the ice thickness. Millan et al. (2019) data set has less bias than the other two; however, the correlation remains low ($R^2$=0.39). In general, previously published estimates are not capable of properly resolving the ice thickness of a lake-calving retreating glacier such as San Quintín.





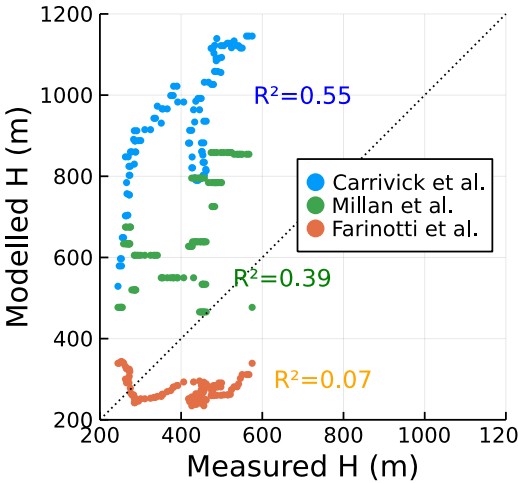

**Figure 9.** Scatter plot of the measured ice thickness of San Quintín glacier vs. previously published estimates of Carrivick et al. (2016), Farinotti et al. (2019), and Millan et al. (2019).

## 4 Discussion

This paper presents an ensemble of indirect measurements that, taken together, confirm the presence of a floating terminus: a flat ice surface of the lowest reaches of the glacier (Fig. 7a), high GPR reflection from the glacier bed, indicating a high water content (Fig. 7b), high relative water depth $\omega$ reaching a flotation criterion of 0.9 (Fig. 7c), and an inversed altitudinal profile

of ice surface elevation change (Fig. 7d). When the glacier front becomes afloat, calving processes increase and switch from small-scale calving of ice pillars and slabs to detachment of large tabular icebergs (Warren et al., 2001; Benn et al., 2007), leading to a much faster retreat of the front. The icebergs produced are of significant size, reaching surface areas higher than $10^4$ m$^2$ (Podgórski and Pętlicki, 2020).

An airborne GPR survey remains one of the very few reliable methods of measuring the ice thickness of Patagonian glaciers

(Millan et al., 2019). Due to the very high water content, significant ice thickness and intense crevassing, temperate outlet glaciers present a challenge for bed detection (Zamora et al., 2009). The application of GPR also allows for insight into the properties of the bed, based on the analysis of the amplitude of the reflected signal (Gades et al., 2000). Although it needs additional data to provide further support for the presence of water, BRP$_r$ remains a useful measure of the water content at the glacier bed (Gacitúa et al., 2015).

Unfortunately, the geometry of the proglacial lake remains unknown, and due to the remoteness of the site and the persistent presence of large tabular icebergs, direct measurements are highly difficult to perform. The regions where the ice front position remained stable for several years while in the other places it was retreating can be suspected to be more shallow, serving as pinning points for the glacier bed. Since the glacier is detached from the bed in the frontal area, the lake bottom there is lower



than the measured minimum elevation of the glacier bed and therefore the water depth reaches more than 200 m. This value is similar to the neighbouring San Rafael glacier lagoon (Koppes et al., 2010), connected to the Pacific Ocean through the canal de Tempanos. The sedimentation rate in the San Quintín glacier lake can be expected to be comparable to that of the San Rafael lagoon, as both glaciers are of similar size and share the same geological setting. Unlike San Rafael, which has a small direct

connection to the ocean and is exposed to tidal circulation, a larger portion of the suspended material released by San Quintín remains trapped in the proglacial lake that has only limited outflow through shallow rivers (Podgórski and Pętlicki, 2020). The bedrock at the new proglacial lake is very likely soft due to the high sedimentation rate taking place in there. The water color at the proglacial lake is typical of a glacier milk, and the river outflowing the lake is reaching the ocean only few km downstream where is leaving a clearly visible sediments plume spreading along the coast.

Interestingly, the current disintegration of San Quintín tongue may serve as an analogue to a possible future retreat of the Malaspina glacier in Alaska, as both share similar characteristics, forming a large piedmont lobe on a flat Pacific coastal foreland. Frontal retreat has significantly accelerated after switching from land-terminating to lake-terminating, which eventually triggered a catastrophic disintegration of the lobe. Furthermore, the collapse of San Quintín may provide insight into the processes that occurred during deglaciation of the large lobes of the former Patagonia Ice Sheet, such as those that formed the

General Carrera, Argentino, or Viedma lakes. With the number of lake-terminating glaciers worldwide increasing due to the widely observed frontal retreat, it is of great importance to fully understand the processes that govern lacustrine calving.

It is appealing to draw a comparison of the disintegration of San Quintín to the lacustrine calving on other Patagonian outlet glaciers. Typically, the water in proglacial lakes in Patagonia is stratified with a significant isothermal surface layer of warm water, produced by strong wind mixing and high absorption of heat from the atmosphere, as well as solar radiation (Sugiyama

et al., 2016). On the contrary, the turbid and cold water layer accumulates near the lake bottom. This stratification enhances melt at the waterline, process that is visible in the so-called notches (Warren and Kirkbride, 2003). The supercooled deep water in these lakes can help the development of an ice foot, and causes buoyant calving or underwater calving (Warren et al., 2001; Sugiyama et al., 2019). Unfortunately, the thermal structure in the proglacial lake at San Quintín remains unknown for two main reasons. First, there are no direct measurements, as those are extremely difficult to obtain due to a challenging logistics,

persistent presence of ice melange and large icebergs. Second, floating ice presents a challenge for remote sensing methods for estimating water temperature. The high sediment cover of some icebergs may lead to an overestimation of water temperature, while the spatial resolution of thermal satellite imagery is too low to reliably capture open water. The persistent presence of ice growlers and icebergs has profound consequences for the thermal structure of the lake, as floating ice should effectively cool the surface water. The indirect indication of the low water temperature in the San Quintín lake is that some of the very large

icebergs persist for months before their disintegration (Podgórski and Pętlicki, 2020) whereas other lakes generally have open water with little floating ice. Furthermore, contrary to other large proglacial lakes in Patagonia, the only significant water input to the San Quintín lagoon is the cold meltwater from ice, and the rain that passed through the englacial and subglacial drainage system.

In contrast to the large outlet glaciers of SPI such as Upsala, Viedma, or Grey where the ice flow is laterally confined by

valley walls in the ablation zones, the San Quintín lobe is laterally unconstrained. Its flow regime changes from compressive to





extensive where the glacier passes the final valley gate, to subsequently change again between km 7 and 8 of the longitudinal profile (Fig. 8) to compressive flow that closes transverse crevasses. In the frontal part, the flow regime changes back to extensive, with opening of longitudinal crevasses, extension of 'fingers', and detachment of long tabular icebergs (Podgórski and Pętlicki, 2020). Drawing a parallel with the Larsen Ice Shelf collapse, it can be expected that if the front retreats further

and the lower compressive flow zone is no longer present, the development of transverse crevasses could result in glacier disintegration, since the entire section of the glacier may collapse (De Angelis and Skvarca, 2003; Hulbe et al., 2008). Such loss of a piedmont lobe buttressing flow might have important consequences for the stability of the entire icefield with an acceleration up-glacier. This would lead to a glacier with highly extensional flow and significant transverse crevassing, similar to the neighbouring San Rafael and to the large outlet glaciers of the SPI. The presence of a retrograde bed (Fig. 4) can lead to

a self-sustained frontal retreat, analogous to the predicted retreat of large outlet glaciers of the West Antarctic Ice Sheet (Rignot et al., 2014) or Greenland Ice Sheet (Catania et al., 2018), forming a deep, cold, and stratified proglacial lake along the glacier valley.

It remains an open question what drives the structural difference between the neighbouring two large outlet glaciers of NPI, San Quintín and San Rafael. Why did the piedmont lobe of San Rafael collapse earlier than San Quintín (Winchester

and Harrison, 1996; Koppes et al., 2011)? Among the possible driving, the effects of climate, topography, tides and water stratification on ice dynamics and frontal ablation can be considered (Post and Motyka, 1995; Bartholomaus et al., 2016). A differentiated influence of climatic factors on both glaciers, very closely located, is less probable (Warren, 1994; Schaefer et al., 2013), especially since San Quintín has a larger ablation area and the San Rafael accumulation area is reaching higher altitudes, up to the summit of Monte San Valentin (4,058 m a.s.l.). The tidal influence on San Rafael dynamics by sea water flowing into

the lagoon through the Canal de Tempanos (Reed, 1988) can be related to fluctuating back stress, and to the impact of warm ocean water on subglacial outflows, increasing frontal ablation by subsurface melt (Motyka et al., 2013).

The largest Patagonian glaciers are calving into fjords and lakes where meltwater is abundant due to the prevalence of temperate ice conditions as described by Warren and Sugden (1993), and confirmed with temperatures measured down to 49 m in the ice core collected by Schwikowski et al. (2013) in the accumulation area of Pio XI glacier. Until recently, no floating

parts were observed in any of these glacier calving glaciers (Warren and Aniya, 1999), suggesting that temperate conditions together with very high ice dynamics at the glacier fronts precluded the retention of floating parts after ungrounded conditions were reached. This was noted many years ago by Meier and Post (1987) who argued that temperate tidewater calving glaciers have no floating termini. However, flotation of Patagonian calving fronts has been observed locally and ephemerally (Rivera et al., 2012; Bown et al., 2019; Sugiyama et al., 2019), but no permanent or semi-permanent floating area has been detected,

indicating that the glacier fronts are actually the grounding lines. In this sense, the San Quintín floating area detected here is unique or very infrequent in Patagonia. In spite of this, the ongoing collapse of this floating area seems to confirm the difficulties of highly dynamic temperate glaciers in retaining ice shelves. As can be seen at San Quintín, the floating tongue persists for long periods, reaching 10 years. In an increasingly warmer climate as expected, the ice thinning will probable accelerate reducing the number of years needed for the ice to reach near flotation conditions. We suspect that the current

floating part detached from the bed around year 2005 when there was a sudden increase in the ice flow velocity of the terminal



part. This hypothesis can be supported by the analysis of an inverted altitudinal profile of ice elevation change already between 2000 (SRTM DEM) and 2011/2014 (TanDEM-X DEM), pointing to a compensation of ice surface elevation change by ice bouyancy. Nonetheless, the disintegration of the floating tongue continues with abrupt frontal retreat episodes.

## 5 Conclusions

We present unique in situ data on the ice thickness of a large Patagonian outlet glacier and analyse satellite remote sensing data that show the collapse of its piedmont lobe, the last in the Andes. The grounding line is located 1200 m from the current front position and 2 km from a possible pinning point, where the glacier leaves a mountain range and enters a flat coastal foreland of the Isthmus of Ofqui. The subglacial topography of San Quintín up-glacier from the lagoon remains unknown, although it can be suspected that the negative elevation can extend over many kilometres, possibly even reaching icefall just below the

plateau (Millan et al., 2019). If San Quintín retreats from its current position, an increase in driving stress caused by surface steepening and, consequently, an acceleration of flow from the accumulation area can be expected. This may potentially lead to destabilisation of the entire icefield, as the San Quintín accumulation zone is constrained by a large subglacial trench shared with the other two major outlet glaciers of the NPI, San Rafael and Colonia. This would have considerable implications for possible deglaciation and future dynamic stability of the entire NPI.

*Data availability.* The GPR profiles are available at Zenodo repository: https://doi.org/10.5281/zenodo.7607535





**Appendix A: DEMs generation from VHR optical stereo imagery using ERDAS IMAGINE**

The orthophoto and DEM generation was done using the IMAGINE Photogrammetry extension to ERDAS IMAGINE (Hexagon Geospatial). It is a semi-automatic tool for generating elevation data from stereo-pair satellite images. Among the supported data are images acquired by the Pléiades and WordView-2 satellites. Using this programme, it was possible to generate DEMs without any ground control points (GCP). The workflow consisted of four steps:

1. Creating a block file

2. Importing a RPC file information

3. Performing a block triangulation

4. Generating a DEM and orthophoto

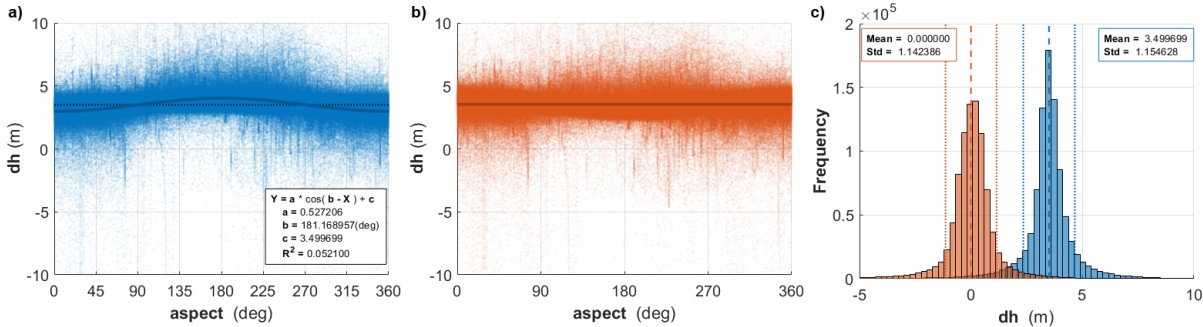

**Figure A1.** Elevation difference (dh) in relation to the aspect: (a) uncorrected; (b) corrected using Eq. A1 without applying the $c$ factor, (c) histograms before and after the correction on stable terrain below 50 m altitude. While both histograms are well-distributed, only the final product is centred around zero.

In the first step, a block file was created, the geometric model (Rational Function) and satellite type, as well as the coordinate system (Geographic Lon. Lat., WGS 84) were selected. The images were then imported, and pyramid layers were created. In the second step, the satellite orientation for both image pairs was calculated using the RPC model (rational polynomial coefficients). It is used to represent the image geometry of the sensor as ground coordinates It is recommended to use Ground Control Points (GCPs) in order to refine the RPC function further. High-precision GCPs were not available for this study because there is no high-resolution DEM or in-situ GPS measurements. First, the tie points were automatically assigned. Then, block triangulation was performed, combining the two separate images. Finally, the DEM was extracted and an orthoimage of each image pair was created. According to Rieg et al. (2018), use of GCPs would improve the absolute vertical, but not necessarily the horizontal accuracy. If both DEMs are well corrected and co-registered to each other, over a stable terrain, the relative change results are still of great value. In this study, the DEMs were corrected using the relationship between the elevation difference of both DEMs and the aspect of the to-be-corrected DEM (Rivera et al., 2005; Nuth and Kääb, 2011).



$X$ and $Y$ shift vectors were calculated and the bias-corrected (see Fig. A1). An elevation-dependent correction would also be possible, but because there are no major elevation differences within the investigated scene, it was not chosen. Furthermore, to optimise the DEM correction, only the area around the lagoon (glacier foreland below an altitude of 50 m a.s.l.) was selected to calculate the shift components. Therefore, no noise related to glacier movement was incorporated into the correction.

The shift was calculated as follows:

$$Y = a \cdot \cos{(b - X)} + c \tag{A1}$$

Where $Y$ is the elevation difference, the length of the shift vector, $b$ the orientation of the vector, $X$ the aspect and $c$ the elevation bias. The 2017 Worldview-2 DEM was shifted 0.5 m towards 181º (south) and raised 3.5 m (see Fig. A1). Aguilar et al. (2013) suggests a vertical accuracy of 2.56 m for a Worldview-2-DEM without GCP, while Podgórski and Pętlicki

(2020) compared the WorldView-2 DEM with TanDEM-X DEM and estimated the vertical precision at 1.70 m. A similar error, if not smaller, can be expected for the Pléiades.

*Author contributions.* MP designed the study, conducted field measurements, made major part of the calculations and analysis, wrote majority of the text; JU set up GPR, JO and JU contributed to GPR data processing, JR produced DEMs, and AR and FB contributed to the analysis and discussion. All authors contributed to the final version of the manuscript.

*Competing interests.* No competing interests are present

*Acknowledgements.* This work was funded with ANID INICIACIÓN 11170937 grant and has been supported by a grant from the Priority Research Area (Anthropocene) under the Strategic Programme Excellence Initiative at Jagiellonian University. The authors are particularly grateful to Philippe Reuter and Suma Air company for their help in conducting the 2019 field campaign, and to Etienne Berthier for his help with Pléiades aquisition. The Pléiades stereo-pair used in this study was provided by the Pléiades Glacier Observatory initiative of the French

Space Agency (CNES) and the Laboratory of Space Geophysical and Oceanographic Studies (LEGOS).





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
