# Peer review of "Frontal collapse of San Quintín glacier (Northern Patagonia Icefield), the last piedmont glacier lobe in the Andes"

_The Cryosphere, 2023_

## Referee Comment (RC1)

**Review of Petlicki et al. The Cryosphere**

Petlicki et al. report possible mechanisms of ice front retreat with of San Quintin Glacier in the northern Patagonian icefield with their new helicopter-borne ice radar survey. The study reports a unique ice radar dataset, although I think the results of the study are not convincing enough to lead the discussion and conclusion. Their ice radar survey is quite limited in space to prove their opinion. Unfortunately, I generally wonder if the quality of the paper is enough to be considered for publishing in the journal. I hope my comments are useful to improve the manuscript.

**General comments**

1. Is it possible to say there is a floating ice tongue and grounding line with your dataset? At least, I've not convinced by their results. Part of the ice front may be afloat indicated by tabular icebergs observed in the lake. However, what we expect by saying floating ice tongue is the entire region of the ice front floats. It looks like the authors' opinions are largely based on their longitudinal radar profile along the centerline of the glacier. Because the centerline of the glacier is expected to have deep bed topography, it is not possible to extend their interpretation to the entire terminus region. Furthermore, if the ice front showed "fingers" of tabular shape, it is rather suggested that the lateral part of the ice front is grounded and caused shear strain to the tear ice front. I do not agree that the entire ice front is floating as you interpreted in Figure 8.

2. I'm not sure how the authors defined the ice-bed interface. It benefits showing the original radargram without the red interpolated line since it looks blurry, especially for the longitudinal profile AA'. Also, what is the difference in ice thickness or bedrock elevation at the crossing point between AA' and BB' and between AA' and CC'?

3. I'm a bit confused with the structure of the paper. I think the most important result of the study is the ice radar survey. I would suggest that the authors first explain their radar survey before the satellite dataset in the method and result sections.

4. I think the authors need to improve their description of satellite image analysis and its results. For example, what is the uncertainty of the annual ice speed obtained from optical satellite imagery? These are the regions where we have many cloudy days, it is fair to mention the number of images used to calculate the annual ice speed with an estimated uncertainty range. Why don't you show a time series of ice front position (or ice extent area) change somewhere in your figure? Figure 3a is not a bit busy and not easy to see to distinguish how the glacier front changes over time.

**Specific comments**

Abstract: Can be more qualitative? For example, what is the maximum ice thickness you observed, how close to flotation, or what is the rate of ice front change?

Abstract L7-8: Can it be more precise about how your study contributes to the study topic? Even after reading your discussion, it was not clear to me how your result gives insight into processes governing the frontal retreat of lake-terminating glaciers.

P1, L7-17: Is there any relevant paper about the lobe?

P3 Figure 1: label a and b is missing in the figure.

P4 L12: 2.1. Satellite imagery,… What are the estimated errors of the satellite-derived dataset used in your study?

P5 L4: m・s^-1… I don't know if it is common to indicate a dot between the units.

Please check the same problem throughout the manuscript and correct it if it is necessary.

P5 L13-14: I wonder how this uncertainty propagates your floatation calculation.

P5 L17, 23: How did you define n1 and n2 for obtaining the bedrock reflective power? Need more descriptions.

P5 L18: At the first glance at Figure 4a, I was not sure where is the ice-rock interface without the red line. How did you obtain the red line to conduct the following analysis?

P7 Fig 2: Why you are not showing data for the longitudinal profile AA'?

P7 Fig 2 caption: …200m… It looks like space is missing between the value and the unit.

P7 L8: What is the overall error of your floatation calculation that arises from the uncertainties in thickness, ice surface elevation, and ice density?

P8 Section 3.1: It may be worth also showing time series of ice front position changes or changes in the terminus ice area. It is not easy to distinguish where the ice front retreat in which year in Figure 3a.

P10 L4: … only slightly negative… But I can also see the most negative thinning occurring near the middle of the terminus.

P10 L4 … with high variability …. It can be more qualitative like from *** to **

P10 L10-11: What is the difference in ice thickness or bedrock elevation at the cross-over point between cross-sectional and longitudinal profiles?

P10 Fig 4: How do you define the red line defined as the ice-bed interface? It looks like the reflection is weak to define the boundary. I wonder what it looks like without the red line. Can you indicate where is the location you have transect profiles on the panel a?

On the lower panels of the BRPr and IRP, there is no legend or description of which plot represents which variables. Add legend and description in the caption.

P11 Fig 5: Is it possible to calculate the ice buoyancy along cross-sectional profiles? It may help interpret your spatial classification of floating, near floating, and grounding in Figure 8. I would use different color codes for panels a and b to avoid confusion. It looks a space is missing in the x label before the unit. The location of the dot over Z looks strange.

P11 Fig 6: You could also show a time series of changes in the ice front position or a time series of the lobe area. I also wonder what is the uncertainty of the annual ice speed you are showing. The number of velocity maps may be significantly different over the year, and you need some more caution to using annual ice speed. Particularly, the glacier shows large seasonal ice speed variations with SAR-derived ice speed.

P13 Fig 7a: Why the topography is not observed between 5 and 6 km from the ice front? How did you calculate ice buoyance for the region without knowing bed topography?

P13 Fig 7c: What the dotted horizontal lines mean? Always better explain it in the caption or legend.

P14 Fig 8: How did you classify those spatial classifications of the close flotation and grounded regions?

P14 Section 3.6: How did you compare your observed thickness with a modeled thickness which has a quite different spatial resolution? Do you have any suggestions to improve the modeling thickness by finding the large discrepancy between observed and modeled ice thicknesses? Also, this section sounds better placed in the Discussion.

P5 Figure 9: There is a typo in the x-tick label: "120" should be "1200". Explain what the black dotted line means.

P15 L10-11: How did your study overcome these challenges? I would appreciate it if you could add some description in the method or introduction.

P15 L16-18: I'm not sure where is the region you are explaining about.

P16 L3-4: I would expect different sedimentation rates by considering the substantial difference in the ice speed between the glaciers (e.g., Koppes et al., 2015 Nature). San Rafael glacier flows an order of magnitude faster than San Quintin glacier (e.g., Mouginot and Rignot 2014 GRL).

P16 L9: …spreading along the coast. You could refer to Figure 1.

P16 L10: Have you consider to compare your results with the previous study published recently (Tober et al., 2023 JGR)?

P16 L17-33: It looks not like this paragraph is logical. In the first sentence, you say that you will compare the disintegration of San Quintin with other Patagonian lake-terminating glaciers. But the following discussions are all about lake properties.

P16 L25: Would it be ice mélange, since it comes from French?

P17 L4: The authors may need more caution to compare the lake-terminating glacier with the ice shelf. Even if San Quinin Glacier has a floating tongue, this is not an ice shelf so you could not simply compare each other. Also, the scale of the glaciers you are comparing is one to two orders of magnitude different.

P18 Conclusions: I think the conclusion is not based on their study and includes many speculations. I would suggest the authors rewrite the conclusions based on their results in a qualitative way.

P18 L6: …grounding line is located… I wonder how you can locate the grounding line with your three thickness survey profiles.

P18 L9: Have you discussed your data with the previous study discussion? It is not common to cite a paper in the conclusion. Because you are not discussed your thickness observation in the previous study, I was a bit surprised by your sudden argument about the potential destabilization of the entire ice field in the following sentences.

---

## Referee Comment (RC2)

Petlicki et al review

Petlicki and others present data from a 2019 airborne ground penetrating radar survey of the San Quintín glacier, Patagonia. The authors do a nice job of summarizing the data and explaining how the glacier has changed over the past ~20 years, mostly from satellite imagery and serval previous studies. While the 2019 airborne survey was rather spatially limited, the authors use remote sensing to broaden their results. The authors present a brief discussion on the future of this piedmont glacier, although the Discussion section could be improved by increasing the number statistical and quantified variables. Many of the calculated variables and figures are difficult to understand and do not enhance the paper as much as they should.

Overall, the paper is scientifically sound but suffers immensely from grammatical and language issues. Much of this manuscript is difficult to review due to incomprehensible run-on sentences and grammatical errors. I recommend major revisions and a rewrite of the Discussion/Conclusion sections that focus on how this study will improve the understanding of the San Quintín glacier and how it compares with other studies in the region. The manuscript is not necessarily novel in technique or scope, but it should at least contribute to the greater understanding of the northern Patagonia icefield if written better.

Other comments:

Abstract:

There's room for more quantification of results and major findings. This abstract is quite short and could use a few additional sentences to invite the reader to continue learning from the rest of the paper.

L1: It's always nice to use the term "anthropogenic climate change," instead of "climate warming," to remind readers (especially in the US) that this problem is manmade.

L2: Sometimes it's "San Quintín" and sometimes it's "San Quintín Glacier." Pick one and be consistent.

L3-4: How "new" is the proglacial lake?

L4: Word "new" is repeated from previous sentence. Consider "We present results from a 2019 airborne GPR survey…"

L4-7: Long run-on sentence. Consider splitting into two.

L6: Can you quantify "shortly" or at least give some estimate of how long it will survive?

Introduction:

L21-L4 (Page2) very long run-on sentence that could be simplified and split into two.

P2 L8: Both Fig. 1 and Figure 1 are used, be consistent.

P2 L12: How large is the net negative surface mass balance?

P2 L18: Quantify "extremely wet and temperate maritime climate."

P2 L21: Explain who Steffen (1900) is and give their full name.

P2 L28-33: Long confusing sentence.

P3: Define SAR interferometry.

Data and Methods:

Need to define GEEDiT, FAU RETREAT, NASA MEaSUREs, ITS_LIVE, etc.

Please address the accuracy of the DEMs (you state 2 m resolution, but how accurate) and ice flow velocities.

P5 L3: I'm not sure if there's a typo in "was set at 3 kHz of the pulse repetition frequency," or what "of" means in this case.

P5 L8: How constant is "electromagnetic wave velocity propagation in ice of 0.168 m·ns−1??" I know the speed should change through different ice densities and depending on how saturated the ice is with lagoon water.

P5 L12: Likewise, please address how constant the airwave speed is. Does this vary with air density, temperature, etc.? Provide a citation. Please address how accurate you are using a constant velocity for both ice and air, and how these numbers compare with overall accuracy of your methods.

P5 L27: How do you know how thick the ice is before conducting the BRP analysis? This seems somewhat circular.

P8 L9: Unclear what "much ice supports the ice column before it reaches flotation" is trying to say.

P11 L2: The relationship might be "clear," but it is quite complex. Some additional annotations on the figure might help?

Results
P8 L18: Retreated over 3 km since when? 1993? And until 2020? This sentence is unclear.

P10 L6: Say that the radio echo sounding is from GPR, otherwise it's unclear what is being talked about.

P10 L10: "It" refers to the glacier bed? Be specific.

P10 L12: State which GPR profile is the longitudinal.

P10 L11-15: Which is the BRPr and which is the IRP? Need to say which is black and which is red.

P11 L9: Here the authors use "up-glacier," while previously it's been "upstream." Be consistent.

P11 L11-12: How consistent? Give some sort of comparison. What advantage does the RETREAT FAU give you having multiple measurements per year?

P12 L3-4: Give a citation for assuming a parabolic profile of the lake.

P12 L11: Please explain the 2011/2014 survey, is this from both years or an average of the two?

P13 L8: Quantify "was very small."

P13 L10: Please quantify this assumption. How much does the ice elevation rate change over 2000-2019 in your data? I agree that this looks somewhat reasonable visually, but quantify it to make sure.

P14 L2: Define "near future" and provide numbers for when each section will disintegrate.

P14 L5-12: Where are the measurements and model values coming from? Where on the glacier are these thicknesses? The reader needs to know how to compare these values.

P14 L8: Quantify correlation coefficients "are very low" and bias "is very high."

Discussion
First several paragraphs seem like "introduction" material instead of discussion
Almost none of the text in this section is appropriate for "discussion," since the results are barely discussed. Please comment on the results presented earlier and why they are important, how they compare with previous studies, and the major implications of this research.

P17 L1: How far "up glacier" is the "final valley gate?"

Conclusion:
This is the first time that you state where the grounding line and possible pinning points are, which should have been discussed earlier.
This conclusion does not recap what is done in the paper, nor does it highlight major findings or future work. The conclusion needs to wrap up the paper nicely and reiterate the purpose of this study.

Figure 1: Reichert and Exploradores glaciers names are a bit hard to read. Contours (presumably in m) should be stated in figure caption. Would be nice to have scale on inset to see how large the lake is.

Figure 3: Amazing figure. The reader is really only interested in the entire extent of the figure for panel a (and even this panel could be zoomed in), otherwise panels b and c should be zoomed in on the current glacier extent. This figure is just so amazing and there's lots of wasted white/lake/land space.

Figure 4: Need to state what the blue and red lines signify, and/or label the air, ice, water/bed. The red line in the bottom figures is hard to distinguish from the red dots. Which is the BRPr and which is the IRP? Need to say which is black and which is red.

Figure 5: Not sure if relative water height needs () as units for panel a. I'd consider moving "BRPr (dB)" and "w ()" into the label for the colorbar instead of the figure legend. The dot above the Z in the x-labels is not centered. I'd recommend replacing "Z" with "elevation" in the second plot.

Figure 6: Ice flow velocity label should be vertical and next to the colorbar. No need to label "year" for either plot.

Figure 7: Great figure. I'd recommend zooming in to 5-11 km for panels b-e, since there is no data for the left half of these figures. I understand wanting to line them up with panel a, but it's difficult to interpret when all the data is squished together in the right half of these figures.

Figure 9: Should be previously published estimates "from." The 1200 value on the x-axis is cropped out.

Data availability: Very straightforward and easy to use data. Nicely done.

Appendix A:
P19 L1: Imagine is repeated.
P19 L7: Define RPC.